# Rescuing neural spike train models from bad MLE

**Diego M. Arribas**[1,2]  **Yuan Zhao**[1]  **Il Memming Park**[1]

[1]Department of Neurobiology and Behavior
Center for Neural Circuit Dynamics
Stony Brook University, NY, USA
[2] Biomedicine Research Institute of Buenos Aires, Argentina
`{diego.arribas,yuan.zhao,memming.park}@stonybrook.edu`

## Abstract

The standard approach to fitting an autoregressive spike train model is to maximize the likelihood for one-step prediction. This maximum likelihood estimation (MLE) often leads to models that perform poorly when generating samples recursively for more than one time step. Moreover, the generated spike trains can fail to capture important features of the data and even show diverging firing rates. To alleviate this, we propose to directly minimize the divergence between neural recorded and model generated spike trains using spike train kernels. We develop a method that stochastically optimizes the maximum mean discrepancy induced by the kernel. Experiments performed on both real and synthetic neural data validate the proposed approach, showing that it leads to well-behaving models. Using different combinations of spike train kernels, we show that we can control the trade-off between different features which is critical for dealing with model-mismatch.

## 1 Introduction

Determining the functional relationship between stimuli and neural responses is a central problem in neuroscience. A standard approach is to build a probabilistic generative model and estimate its parameters by maximizing the likelihood of the data under the model. This framework has been applied in diverse scenarios describing the activity of single neurons and coupled populations, extracting low dimensional latent dynamics underlying the data and decoding stimuli that produces neural activity. The extracted model parameters are useful as they allow one to gain insights on the relationship between the observed neural activity, its covariates and the intrinsic dynamics.

However, maximum likelihood estimation (MLE) focuses on making the data likely under the assumed model without really assessing the behavior of the actual samples that the model generates. MLE often leads to models that are unstable, operate at unphysiological regimes or generate samples that fail to capture relevant features of the data. This harms model interpretation and it is a big drawback if the obtained model is intended to be used in simulations.

In machine learning, big improvements in generative modeling were achieved when alternative approaches leading to different loss functions were considered. In their original formulation, Generative Adversarial Networks [1] minimize an approximation to the Jensen-Shannon divergence by training a discriminator model that evaluates sample quality. More recent works have proposed to minimize other loss functions such as the Wasserstein distance [2] and the Maximum Mean Discrepancy (MMD) [3–5].

While these works have focused on the use of deep neural networks to generate synthetic images, models in neuroscience are usually autoregressive and they emphasize interpretability. Here we propose to complement likelihood based approaches with MMD minimization for the autoregressive models that are typically used in neuroscience. Using spike train kernels, MMD can evaluate different

similarity measures between the model-generated samples and the data. We show that the framework is flexible and can be adapted to find models that capture different features of the data.

## 2 Approach

### 2.1 Problem Statement – autoregressive models, MLE, and data generation

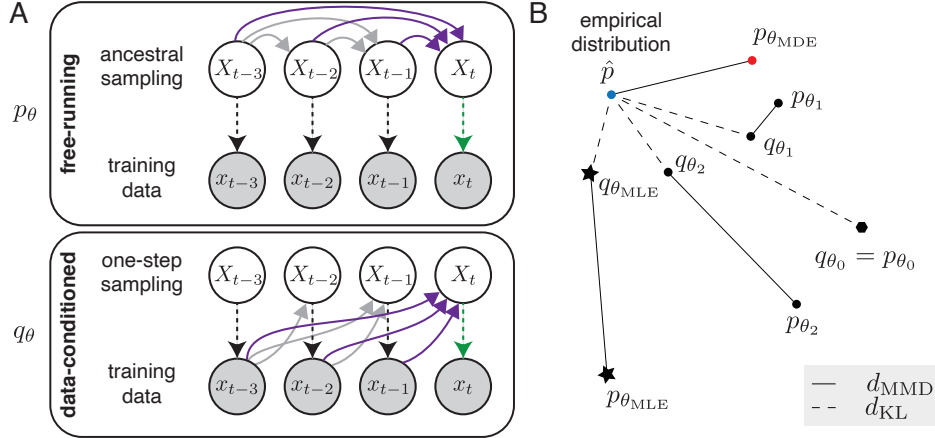

Figure 1: **A) Two distinct likelihoods.** The free-running corresponds to the joint distribution of the probabilistic model. Solid line denotes conditional dependence and dashed line denotes evaluation of the likelihood. The data-conditioned likelihood always conditions on the actual observed past to predict the next time step. **B) Cartoon depicting relative closeness of distributions.** The data-conditioned likelihood $q_\theta$ is optimized to obtain the maximum likelihood estimate $\theta_{\text{MLE}}$ which has minimum KL-divergence wrt the empirical distribution $\hat{p}$, but can lead to a model with unrealistic free-running behavior $p_{\theta_{\text{MLE}}}$. Note that $d_{\text{KL}}$ is not a metric. See Sec. 2.3 for details.

We denote by $x_t$ an observation at time $t$ and by $X_t$ the corresponding random variable in the stochastic process. Given a sequence of observations or time series, $\{x_t\}_{t=1}^T$, a general autoregressive model predicts $X_t$ based on the past $X_{<t}$, which can be concisely encapsulated as the conditional probability $p(X_t|X_{<t}, u_{<t}, \theta)$, where $\theta$ denotes the model parameters and $u_{<t}$ are optional (observed) covariates. The standard maximum likelihood estimation (MLE) procedure yields the parameters that maximizes the likelihood for predicting this one-step prediction,

$$\theta_{\text{MLE}} = \operatorname*{argmax}_{\theta} \prod_t p(X_t = x_t|X_{<t} = x_{<t}, u_{<t}; \theta). \tag{1}$$

The disparity between the one-step prediction and longer-term forecasting is common in many autoregressive models, including recurrent neural networks (RNNs) [6–9]. For example, sequences freely generated from natural language models trained to predict the next token typicaly exhibit over-representation of unnaturally long sequences. A fundamental issue common to these problems is the difference between the free-running joint distribution and the data-conditioned joint predictive distributions (Fig. 1A):

$$p(X_t, X_{t+1}, \dots X_{t+p}|x_{<t}; \theta) = \prod_{s=0}^{s=p} p(X_{t+s}|X_{t:t+s-1}, x_{<t}; \theta) \qquad \text{(free-running)} \tag{2}$$

$$q(X_t, X_{t+1}, \dots X_{t+p}|x_{<t+p}; \theta) := \prod_{s=0}^{s=p} p(X_{t+s}|x_{<t+s}; \theta) \qquad \text{(data-conditioned)} \tag{3}$$

where $X_{t:t+s-1}$ denotes $\{X_t, X_{t+1}, \dots, X_{t+s-1}\}$. In Eq. (2), the autoregressive model's joint distribution is conditioned on its own prediction after time $t$ while in Eq. (3) the conditioning is always done on the observations as in typical MLE (Eq. (1)). This conditioning of data is similar to the *teacher forcing* in the context of recurrent neural networks [10].

For example, for an autoregressive dependency that induces self-excitation, (2) and (3) can behave vastly differently—assuming the observed data are from a stable system, conditioning on the observed

history (3) produces stable one-step predictions. However, for the same parameters, if one sampled trajectories using ancestral sampling from (2), runaway self-excitation might be generated. To illustrate this, consider one of the simplest autoregressive models, the *linear* autoregressive model of order $p$, AR($p$):

$$X_t = \sum_{\tau=1}^{p} a_\tau X_{t-\tau} + \epsilon_t \tag{4}$$

where $\{\epsilon_t\}_t$ are white Gaussian noise, and $\theta = \{a_1, \ldots, a_p, \text{var}(\epsilon_t)\}$. For AR($p$) models, the MLE could result in unstable parameter regime of self-amplification when the poles of the linear system are close to the unit circle [11, Ch. 10]. Fortunately, the condition of stability is exactly known for AR($p$) models, and an estimator that constrains parameters to lie within the stable regime has been developed [11, Ch. 10]. For instance, for AR(1), $|a_1| < 1$ guarantees stability and stationarity, while $|a_1| > 1$ guarantees instability.

However, beyond those *linear* models, the intractability of the free-running distribution makes it difficult to directly optimize for it. For instance, the autoregressive point process models, often referred to as the generalized linear models (GLMs) in neuroscience [12, 13], suffer from the issue of instability as well:

$$X_t \sim \text{Poisson}\left(\lambda(X_{<t}, u_{<t}; \theta)\right) \tag{5}$$

$$\lambda(X_{<t}, u_{<t}; \theta) = \exp\left(\sum_{\tau=1} h_\tau X_{t-\tau} + \sum_{\tau=1} a_\tau u_{t-\tau} + b\right) \tag{6}$$

where $\theta = \{\{h_\tau\}_\tau, \{a_\tau\}_\tau, b\}$ are the parameters, $\{h_\tau\}_\tau$ and $\{a_\tau\}_\tau$ are referred to as the *history filter* and *stimulus filter*, respectively, and $b \in \mathbb{R}$ is the bias.

Self-excitation in the inferred history filter has been observed on both short and long time scales, with different proposed causes: bistable or overdispersed data [14], periodic or bursting patterns [15], omitted covariates [16], ramping firing rates [17], and lack of data [18]. These history filters have the potential to generate runaway self-excitation. Recently, a number of approaches have attempted a resolution for the point process GLM and related models: Gerhard et al. [19] approximate the free-running distribution over the history using a quasi-renewal process approximation which is later extended by Chen et al. [18]. Rule and Sanguinetti [20] also provide an approximation of this distribution using moment matching. On the other hand, Hocker and Park [21] use Gibbs sampling to obtain the marginalized free-running likelihood for multi-step prediction which is computationally prohibitive. In this paper, we take a fresh stab at this problem.

## 2.2 Goodness-of-fit measures

To complicate the matter, the goodness-of-fit measures often assume the data-conditioned likelihood. As a result, the standard log-likelihood based measures such as deviance, information, or pseudo-$r^2$ [13] as well as the quantification of interval distribution using the time-rescaling theorem [22] fail to reliably predict whether the fit model would generate free-running samples similar to the data [19, 21]. Here we discuss various forms of goodness-of-fit measures for GLM-like models which could be also useful for fitting.

A statistical divergence is a non-negative function that quantifies how dissimilar two distributions are [23], thus it can be used as a goodness-of-fit measure: the smaller the divergence, the better the fit. Consider the Kullback-Leibler (KL) divergence $d_{KL}(\hat{p} \parallel q(\theta)) := E_{X \sim \hat{p}}\left[\log \frac{\hat{p}(X)}{q(X|\theta)}\right]$ between the empirical data distribution $\hat{p}$ and the data-conditioned likelihood $q(\theta)$ from (3), where $\hat{p}(\{X_t\}_t) = \prod_t \delta(X_t - x_t)$. In this context, the standard MLE is equivalent to minimizing the KL divergence, that is,

$$\theta_{\text{MLE}} = \underset{\theta}{\text{argmin}} \, d_{KL}(\hat{p} \parallel q(\theta)) = \underset{\theta}{\text{argmax}} \, q(\{x_t\}_t \mid \{x_t\}_t, \theta) = \underset{\theta}{\text{argmax}} \prod_t p(x_t \mid x_{<t}, \theta). \tag{7}$$

As our goal is to find a model that can generate time series that resemble the data, it naturally leads to the following minimum divergence estimation (MDE):

$$\theta_{\text{MDE}} = \underset{\theta}{\text{argmin}} \, d(\hat{p} \parallel p(\theta)) \tag{8}$$

where $d$ is a divergence and $p$ is the free-running likelihood of (2). From this first principle, our challenge is finding a divergence such that the optimization (8) is computationally feasible. Unlike in the standard MLE, using KL divergence generally results in an intractable objective function.

We investigate a widely used kernel-induced statistic called maximum mean discrepancy (MMD) [24]. Given a positive definite kernel $k$, we can embed a probability measure $p$ in the corresponding reproducing kernel Hilbert space $\mathscr{H}$, i.e., $p \mapsto p_k := \int k(\cdot, x) \, dp(x) \in \mathscr{H}$. MMD measures the distance between the kernel embeddings,

$$d_{\text{MMD}}(p, q) = \|p_k - q_k\|_{\mathscr{H}} = \sup_{f \in \mathcal{F}} \left( \underset{X \sim p}{\text{E}}[f(X)] - \underset{X \sim q}{\text{E}}[f(X)] \right) \tag{9}$$

where $\mathcal{F}$ is a unit-ball in $\mathscr{H}$ [24]. To find parameters $\theta$ that generate free-running samples similar to the data, we can minimize $d_{\text{MMD}}(\hat{p}, p(\theta))$, the MMD between the empirical data distribution $\hat{p}$ and the free-running distribution $p(\theta)$. Depending on the choice of kernel, MMD differentially weighs the features in the input space. Hence, MMD can be tuned to produce goodness-of-fit statistics sensitive to that of the inducing kernel. Importantly, if the kernel $k$ is *characteristic*, the induced MMD is a divergence (and a metric), i.e., it vanishes if and only if the two distributions are identical [25].

## 2.3 MMD Professor Forcing

Given parameters $\theta$ and a time series $\{x_t\}_t$, if the free-running (2) and data-conditioned (3) distributions agree, we might expect $p(\theta)$ to be a good description of $\hat{p}$. If the data is stable, the free-running behavior of the model won't diverge from the data-conditioned behavior. This is the idea introduced by Professor Forcing [9] where they encouraged the behavior generated by a RNN in free-running and teacher forcing modes to be the same. Then, MMD based goodness-of-fit that measure the closeness between $q(\theta)$ and $p(\theta)$ can be used. Note that for autoregressive models, the agreement between the free-running and data-conditioned distributions may be trivially achieved if the model does not depend on the history (e.g., Poisson process). Thus these MMDs cannot be optimized on its own. Fig. 1B illustrates the ideas here with various possible model parameters and their corresponding conditional distributions. While both $q_{\theta_1}$ and $q_{\theta_2}$ are both only slightly worse than $q_{\theta_{\text{MLE}}}$ in explaining data, their corresponding free-running distributions can be very dissimilar (e.g., $p_{\theta_2}$ and $p_{\theta_{\text{MLE}}}$) or very close (e.g., $p_{\theta_1}$). Informally, if both $d_{\text{KL}}(\hat{p}, q_\theta)$ and $d_{\text{MMD}}(q_\theta, p_\theta)$ are small, we might expect $d(\hat{p}, p_\theta)$ to be small too, leading to a faithful generative model.

## 3 Minimizing empirical MMD

Given a kernel $k$ and its associated feature map $\phi : \mathscr{X} \mapsto \mathscr{H}$, using the kernel trick $\langle \phi(X), \phi(X') \rangle_{\mathscr{H}} = k(X, X')$, we can write the (squared) MMD between the empirical data distribution $\hat{p}$ and the free-running distribution $p$ on $\mathscr{X}$ as,

$$d_{\text{MMD}}(\hat{p}, p)^2 = \| \underset{X \sim \hat{p}}{\text{E}}[\phi(X)] - \underset{X' \sim p}{\text{E}}[\phi(X')] \|_{\mathscr{H}}^2 \tag{10}$$

$$= \underset{X, X' \sim \hat{p}}{\text{E}}[k(X, X')] + \underset{X, X' \sim p}{\text{E}}[k(X, X')] - 2 \underset{X \sim \hat{p}, X' \sim p}{\text{E}}[k(X, X')]. \tag{11}$$

Minimizing MMD is then equivalent to minimizing the difference in the statistics represented by $\phi$ between the two distributions. Given $N$ samples $x$ drawn from the data $\hat{p}$ and $M$ samples $x'$ drawn from the free-running distribution $p$, an unbiased empirical estimator of MMD squared is

$$\hat{d}_{\text{MMD}}(\hat{p}, p)^2 = \sum_{\substack{i=1 \\ i \neq j}}^{N} \sum_{j=1}^{N} \frac{k(x_i, x_j)}{N(N-1)} + \sum_{\substack{i=1 \\ i \neq j}}^{M} \sum_{j=1}^{M} \frac{k(x'_i, x'_j)}{M(M-1)} - 2 \sum_{i=1}^{N} \sum_{j=1}^{M} \frac{k(x_i, x'_j)}{NM}. \tag{12}$$

A biased estimator of MMD squared is described in the Appendix. Unlike for MLE, $\hat{d}_{\text{MMD}}$ involves generating samples from the model and measuring their similarity to the data. We propose to minimize MMD by gradient descent on the model parameters. We provide two different variants that rely on different assumptions on the kernel used, and result in different bounds on the variance of the MMD gradient estimator.

### 3.1 Score function estimator of squared MMD's gradient

In general, the only dependence of MMD on the model parameters is through the expectations over the model's samples in Eq. (11). For models with tractable likelihood, given a sample $x'$ we can evaluate $p(x'; \theta)$. We can then rewrite squared MMD's gradient as

$$\nabla_\theta \, d_{\mathrm{MMD}}(\hat{p}, p)^2 = 2E_{x, x' \sim p}[\nabla_\theta \log p(x'; \theta) k(x, x')] - 2E_{x \sim \hat{p}, x' \sim p}[\nabla_\theta \log p(x'; \theta) k(x, x')]. \tag{13}$$

where $\nabla_\theta \log p(x'; \theta)$ is known as the score function [26] (derivation in the Appendix). Given $N$ samples $x$ from $\hat{p}$ and $M$ samples $x'$ from $p$, we can compute a stochastic empirical estimate of the gradient following

$$\nabla_\theta \, \hat{d}_{\mathrm{MMD}}(\hat{p}, p)^2 = 2 \sum_{\substack{i=1 \\ i \neq j}}^{M} \sum_{j=1}^{M} \frac{\nabla_\theta [\log p(x'_j; \theta)] k(x'_i, x'_j)}{M(M-1)} - 2 \sum_{i=1}^{N} \sum_{j=1}^{M} \frac{\nabla_\theta [\log p(x'_j; \theta)] k(x_i, x'_j)}{NM} \tag{14}$$

This is the score function estimator of squared MMD's gradient. In principle, this procedure can be used to minimize MMD for arbitrary kernels in the space of spike trains [27].

### 3.2 MMD Professor Forcing with model based kernels

Although the estimator in Eq.(14) is general, score function estimators tend to have large variance [26]. Here we introduce a different variant of MMD that helps alleviate this problem. We will encourage a model's free-running dynamics to match its data conditioned dynamics by using feature maps (kernels) derived from the model we want to fit. Explicitly, given a model with parameters $\theta$, we can define a feature map $\phi : x \mapsto \nu(\theta) \in \ell_2$ introducing an explicit dependence on the model parameters in the feature map and therefore in $d_{\mathrm{MMD}}$. Hence, minimizing the induced MMD is equivalent to matching the model's behavior conditioned on samples from either distribution.

We illustrate this idea with an example using the autoregressive GLM of Eqs. (5), 6. Given a spike train $x$ and parameters $\theta$, the GLM assigns a time dependent conditional intensity (CI) $\lambda(x, \theta)$ given by Eq. 6. If the CI conditioned on the GLM's free-running spike trains is similar to the CI conditioned on the data, we might expect the GLM's free-running dynamics to better match the data overall. A natural choice of *model based* kernel to achieve this could be $k(X, X'; \theta) = \langle \lambda(X; \theta), \lambda(X'; \theta) \rangle = \sum_t \lambda_t(X_{<t}; \theta) \lambda_t(X'_{<t}; \theta)$ resulting in

$$d_{\mathrm{MMD}}(\hat{p}, p)^2 = \| \mathop{\mathrm{E}}_{X \sim \hat{p}}[\lambda(X; \theta)] - \mathop{\mathrm{E}}_{X' \sim p}[\lambda(X'; \theta)] \|^2 \tag{15}$$

where the squared norm summation is over time.

In general, for any kernel that is a continuous function of $\theta$ [26], we can compute the induced MMD squared with Eq. 11 and its gradient with

$$\nabla_\theta \, \hat{d}_{\mathrm{MMD}}(\hat{p}, p)^2 = \sum_{\substack{i=1 \\ i \neq j}}^{N} \sum_{j=1}^{N} \frac{\nabla_\theta \, k(x_i, x_j; \theta)}{N(N-1)} + \sum_{\substack{i=1 \\ i \neq j}}^{M} \sum_{j=1}^{M} \frac{\nabla_\theta \, k(x'_i, x'_j; \theta)}{M(M-1)} - 2 \sum_{i=1}^{N} \sum_{j=1}^{M} \frac{\nabla_\theta \, k(x_i, x'_j; \theta)}{NM} \tag{16}$$

where we consider the samples generated from the model fixed. [1] Model based kernels introduce an explicit dependence on the model parameters, making the optimization procedure more robust and improving convergence. Model based MMDs are not characteristic in general; The statistics they can match will always be limited by the used model's capacity and zero MMD might be achieved with trivial $\theta$. Therefore, these model based MMDs have to be jointly optimized with the likelihood (see section 2.3).

## 4 Experiments

We demonstrate in the following experiments that it is possible to minimize MMD as described before in both simulated and real data.The following link contains the code used to fit the models https://github.com/diegoarri91/mmd-glm.

## 4.1 Toy example: Learning an autoregressive GLM with MMD minimization using a characteristic kernel

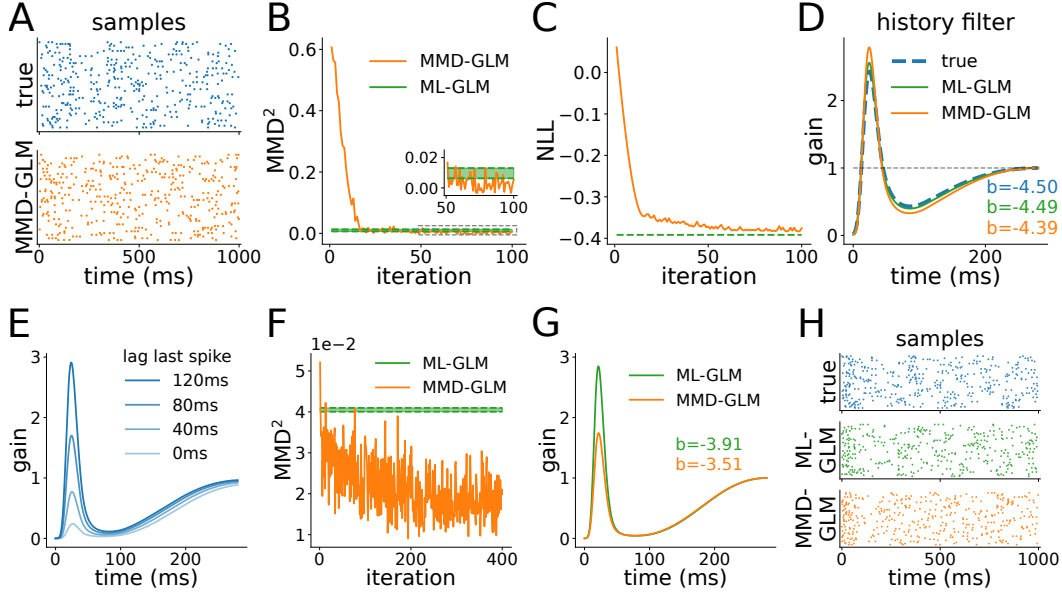

Figure 2: **Learning a GLM's parameters by minimizing MMD. (A-D) without model mismatch, MLE and MMD minimization agree.** A) Samples used for ML and MMD training (top) and samples drawn from MMD optimized GLM (bottom). B) MMD estimates during minimization. C) Negative log-likelihood relative to a homogeneous Poisson process with the same data rate during training. D) History filter and bias used to generate the data and inferred by MLE and MMD minimization. **(E-H) with model mismatch, MLE and MMD minimization can disagree.** E) GQM gain dependence with lag to the previous spike. F) MMD estimates during minimization. G) History filter and bias inferred by MLE and MMD minimization. H) Samples generated from the GQM and used for MLE and MMD minimization (top). Samples drawn from ML-GLM (middle) and MMD optimized GLM (bottom).

To illustrate our formulation we will start by showing we can recover true GLM parameters by directly minimizing MMD without resorting to MLE. We drew 50 spike train samples from an autoregressive GLM (Eqs (6)) consisting of a bias and history filter using Bernoulli noise (Fig. 2A). In this example, we use the kernel

$$k(x, x') = \exp\left(-\frac{1}{\sigma}\int_0^T (I_x(t) - I_{x'}(t))^2 dt\right) \tag{17}$$

where $I_x(t) = \sum_j \Theta(t - t_j^x)$ is the cumulative spike count with $\Theta(t)$ the Heaviside function and $t_j^x$ the spike times of the sample. This is an example of a characteristic kernel [27] so with sufficient data we expect to find the true GLM parameters if and only if MMD is 0. At each optimization step, we draw 200 samples from the model and update the parameters by computing $MMD^2$ and its gradient following Eqs. (12) and (14). $MMD^2$ converges to values around 0 (Fig. 2B) and we recover accurate estimates of the true bias $b$ and history filter (Fig. 2D). We note that the negative log-likelihood decreases during the optimization although it is not being used (Fig. 2C). While MLE also retrieves an accurate estimate of the true parameters, the solution found by the two procedures is slightly different due to finite number of samples. $MMD^2$ is slightly above 0 for the ML-GLM while the likelihood is slightly smaller for the MMD-GLM. In a real application, where there is model mismatch or a non-characteristic kernel is used, the parameters found by MLE and MMD minimization will not be the same. We illustrate this by simulating a model mismatch (Figs. 2E-H). Here we sample from a GQM [28] (Fig. 2H) in which the spike-evoked gain modulation depends on the lag between the current and the last spikes (Fig. 2E). As a GLM can't capture this dependency, $MMD^2$ induced by the characteristic kernel of Eq. (17) is above 0 for MLE (Fig. 2F). By initializing a GLM at the ML estimated parameters and minimizing MMD further we obtain different parameters

(Fig. 2G). This toy example illustrates that in general MLE and MMD minimization are different and will yield different models.

## 4.2 Stabilizing GLMs in real data

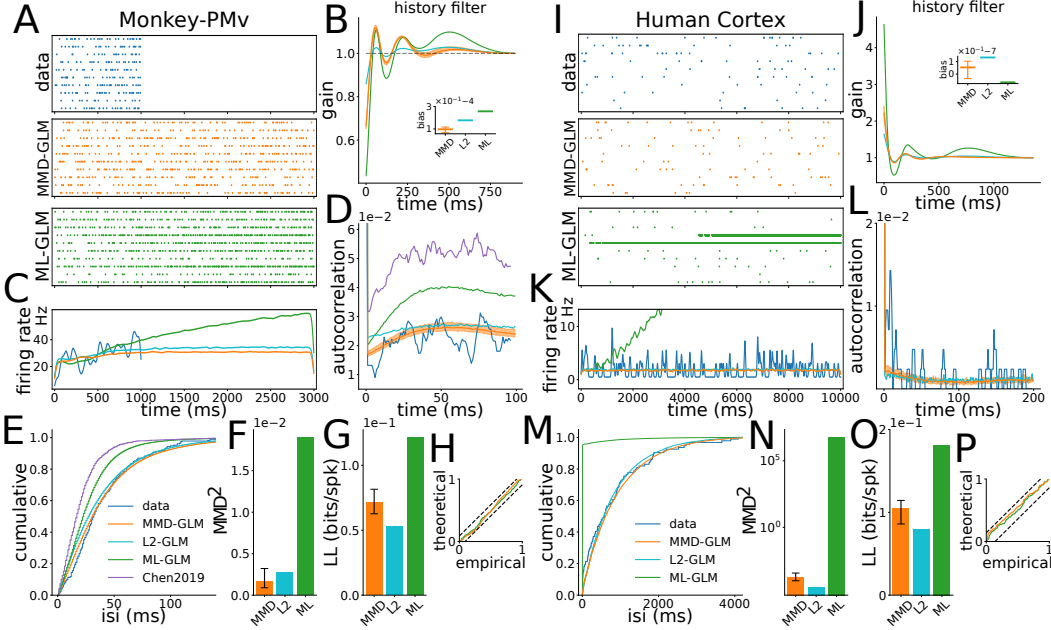

Figure 3: **Multi-objective optimization that combines the likelihood and a model based MMD infers superior generative models.** Data used in [19]. A) Data and generated samples from the models. B) Inferred history filters. C) Smoothed firing rate of the samples from A. D) Autocorrelation of the data (smoothed) and the samples generated from the models. Purple trace obtained using code from [18]. E) Interspike interval distribution of the data and the samples generated from the models. F) Squared MMD estimates. G) Log-likelihoods per spike relative to a homogeneous Poisson process with the same rate. H) Time-rescaled interspike interval distributions. (I-P) same for Human cortex data.

We show here how our procedure can be used to encourage stable GLM parameters that don't suffer from runaway self-excitation. We used two small datasets from monkey ventral premotor cortex (Monkey PMv Figs 3A-H) and human neocortex (Human Cortex Figs. 3I-P) that are prone to yield unstable ML parameters [18, 19]. For both datasets, the ML-GLM showed diverging firing rates (Figs. 3 A,C,I,K). However, for point process GLMs, MLE is a convex problem and it is not computationally expensive. So to leverage its strengths, we initialized our optimizations at the ML estimates and minimized NLL + $\alpha$MMD i.e. the negative log-likelihood plus an MMD penalty term that acts as a regularizer. To determine the weight of the MMD term ($\alpha$) we tried values on a grid and used the smallest $\alpha$ for which the MMD-GLM samples matched the data firing rate within a 10% interval. To study the variability of the stochastic optimization, we repeated the procedure 20 times and report average values. Error bars and shaded regions represent the minimum and maximum values obtained from the repetitions. For comparison, we also show results for L2 regularization on the history filter coefficients where the weight of the regularization was determined in the same way. Although different kernels could be used to stabilize the firing rate, we also tried to improve the samples autocorrelation by using the model based kernel

$$k(x, x'; \theta) = \sum_\tau C_{H_x}(\tau) C_{H_{x'}}(\tau) \tag{18}$$

where $\theta = \{h_\tau\}_\tau$, $C_{H_x}(\tau) = \sum_{t=1}^\tau H_x(t; h) H_x(t + \tau; h)$ and $H_x(t; h) = \sum_\tau h_\tau x_{t-\tau}$. This kernel measures the similarity between the autocorrelations of two history filter convolved spike trains $H_x(t; h)$ and $H_{x'}(t; h)$. Therefore, the optimization encouraged the autocorrelations of the data conditioned and free-running $H_t$ to match.

For both datasets, the MMD-GLM and the L2-GLM produced samples with stable firing rates (Figs. 3A,C,I,K). The MMD-GLM achieved this with less likelihood penalization than the L2-GLM (Figs 3G,O) While the history filters obtained with the MMD-GLM have preferentially reduced late self excitation, L2 regularization history filters are reduced overall (Figs. 3B,J). $MMD^2$ are greatly reduced for both models and datasets. We note that for the Human Cortex dataset, the (model-based) $MMD^2$ is smaller for the L2-GLM than for the MMD-GLM showing that, as discussed, model-based MMDs cannot be used as isolated goodness-of-fit measures. Samples drawn from the MMD-GLM capture the ISI distribution and the autocorrelation of the data (Figs. 3D,E,H,L,M,P).

## 4.3 Capturing different features in the data

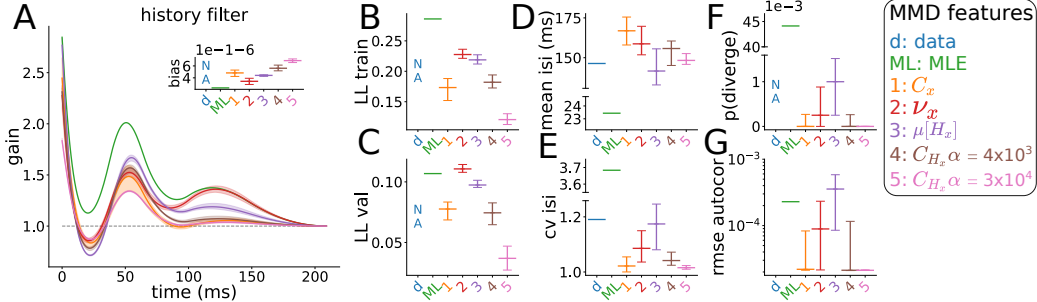

Figure 4: **The use of different kernels leads to different optimized parameters**. A) History filter and bias of the MLE and MMD optimized models. B,C) Training and validation log-likelihoods relative to a homogeneous Poisson process with the same rate. D,E) Mean and coefficient of variation of the interspike interval distributions. F) Probability of a sample showing runaway self-excitation. G) Root mean squared error between the spike autocorrelations of the free-running and validation samples.

In our last experiment, we explored how the choice of kernel influences the resulting model. We used a dataset recorded from the lateral intraparietal (LIP) area of a monkey during a perceptual decision-making task [17, 21]. We used 50 trials for training the models and 50 for validating. We initialized the coefficients of the history filter at zero and the bias at its MLE value for every optimization. We then minimized NLL + $\alpha$MMD drawing 100 trials from the model at each optimization step to compute MMD and its gradient. Fig. 4 shows the selected model parameters we obtained for different kernels (Fig. 4A) together with log-likelihood values (Figs. 4B,C) and different statistics obtained from spike train samples (Figs.4D-G). We computed the mean and coefficient of variation of the interspike interval distribution (Figs.4D,E), the probability of a sample showing runaway self-excitation (Fig.4F) and the error in the autocorrelation (Fig.4G). We considered a sample to suffer from runaway self-excitation if its mean firing rate was greater than 3 times the maximum firing rate over the samples in the data. These statistics were computed by generating 8000 samples for each optimization and using the validation set for the data. To study the variability in the procedure, each optimization was repeated 20 times with the same hyperparameters for each kernel choice. We show the variability over repeated optimizations, reporting medians, minima and maxima of the 20 repetitions.

Models 1 and 2 are independent of the GLM parameters and were optimized using the score function estimator of section 3.1. Model 1 uses the feature map $\phi(x) = C_x$, a sample's autocorrelation while model 2 feature map is $\phi(x) = \nu_x$, a gaussian smoothing of the spike train. Models 3, 4 and 5 were optimized using model-based kernels following section 3.2. Model 3 uses as feature map the mean over time of the history term $\mu[H_t]$. Models 4 and 5 were obtained using a different MMD weight $\alpha$ with the kernel from Eq. 18, which uses the temporal autocorrelation of the GLM's history term.

MLE achieves high likelihood values for both training and validation sets (Figs. 4B,C) but fails to capture any other features in the data due to self-excitation that causes diverging firing rates in 4.4% of trials (Fig. 4F). As expected, all models penalize training log-likelihood to improve their respective MMD (Fig.4B). However, model 2 has greater validation log-likelihood than MLE (4C) indicating that the MMD penalty term can act as a regularizer, helping overfitting. The MMD optimized models captured the mean and cv of the interspike interval distribution with different degrees of accuracy (Figs.4D,E). All MMD optimizations improved model stability over MLE showing very

low probabilities of runaway self-excitation (Fig.4F). Model 3 showed the worst stability with a median value over optimizations of 0.1% of samples showing runaway self-excitation. Even this low proportion of diverging samples has a big effect on the mean autocorrelation (Fig.4G). Although model 4 has a very small probability of generating a diverging sample, model 5 shows that increasing MMD's weight $\alpha$ further reduces this probability at the expense of penalizing other statistics. For all optimizations, all the samples drawn from model 5 were stable. This indicates again that the framework can be used to encourage stable parameters using $\alpha$ to control the trade-off between the different features.

## 5    Discussion

Taking ideas from generative modeling in machine learning, we propose to minimize alternative objective functions to the likelihood as a way to improve sample quality of neural generative models. Here we focused on formulating the framework and exploring the use of different kernels while limiting ourselves to a single model. However, the ideas exposed here can be easily applied to any autoregressive generative model and potentially the benefits could be bigger for more complex models than the point process GLM. In neuroscience, the framework can be easily applied to coupled GLMs, rate and spiking neural networks and dimensionality reduction models.

The main limitation of our proposal is the need to sample and compute kernel similarity during the optimization procedure. Computation can be reduced in many ways. As we did here, MLE can be used for parameter initialization and optimization objectives that jointly use MMD and the likelihood may accelerate convergence. For the score function estimator of MMD's gradient, there are available methods to control its variance and potentially accelerate convergence.

## Broader Impact

Bridging the gap between statistical neuroscientific models such as autoregressive point processes and dynamical systems is a substantial challenge not only from the perspective of generative modelling but also in terms of allowing a dynamical interpretation, that carries with it all the niceties that are afforded by stochastic dynamical systems. As such, while the motivation we drew up on comes from neuroscience, modelling, simulating and analyzing point process dynamics has a broad applicability to biological sciences and other fields.Our method has potential use in modelling within social sciences, geophysics (e.g. earthquakes), astrophysics and finance. In many of those areas stable inference and simulation of future events would directly enable the ability to discern and shape social and economic trends, or effect policy safeguarding against baleful events.

## Acknowledgments and Disclosure of Funding

This work is supported by NSF CAREER IIS-1845836 and IIS-1734910.

## Footnotes

[1] In general, the full gradient involves extra score function terms as the ones in Eq. 14 (see [26])

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
