[Supplementary Material]

# Appendix

## MMD biased estimator

Eq. 12 provides an unbiased empirical estimator of MMD. This estimator requires computing the non-diagonal elements of the Gramian of all the samples (i.e. all possible $k(x, x')$ with $x \neq x'$) which time complexity scales quadratically with the number of samples. If the feature map $\phi$ can be defined explicitly, a biased estimator of MMD squared is

$$\hat{d}_{\text{MMD}}(\hat{p}, p)^2 = \left\| \frac{1}{N} \sum_{i=1}^{N} \phi(x_i) - \frac{1}{M} \sum_{j=1}^{M} \phi(x'_j) \right\|_{\mathcal{H}}^2 .$$

This estimator time complexity scales linearly with the number of samples.

## Derivation of the score function estimator of MMD's gradient

We need to compute

$$\nabla_\theta \, d_{\text{MMD}}(\hat{p}, p)^2 = \nabla_\theta \, \mathbb{E}_{x,x' \sim p}[k(x, x')] - 2\nabla_\theta \, \mathbb{E}_{x \sim \hat{p}, x' \sim p}[k(x, x')].$$

where the dependence on $\theta$ is in the expectations over $p(\theta)$. The log-derivative trick allows as to rewrite the gradient of $\nabla_\theta \, \mathbb{E}_{x \sim p(\theta)}[f(x)]$ as

$$\nabla_\theta \, E_{x \sim p(\theta)}[f(x)] = \int_x \nabla_\theta \, p(x; \theta) f(x) \, dx$$
$$= \int_x p(x; \theta) \nabla_\theta \, \log p(x; \theta) f(x) \, dx = E_{x \sim p(\theta)}[\nabla_\theta \log p(x; \theta) f(x)].$$

Then

$$\nabla_\theta \, \mathbb{E}_{x,x' \sim p}[k(x, x')] = \mathbb{E}_{x,x' \sim p}\left[ (\nabla_\theta \log p(x; \theta) + \nabla_\theta \log p(x'; \theta)) k(x, x') \right]$$
$$= 2 \, \mathbb{E}_{x,x' \sim p}\left[ \nabla_\theta \log p(x'; \theta) k(x, x') \right]$$

and

$$\nabla_\theta E_{x \sim \hat{p}, x' \sim p}[k(x, x')] = E_{x \sim \hat{p}, x' \sim p}[\nabla_\theta \log p(x'; \theta) k(x, x')].$$

Finally,

$$\nabla_\theta \, d_{\text{MMD}}(\hat{p}, p)^2 = 2 \, \mathbb{E}_{x,x' \sim p}[\nabla_\theta \log p(x'; \theta) k(x, x')] - 2E_{x \sim \hat{p}, x' \sim p}[\nabla_\theta \log p(x'; \theta) k(x, x')].$$