[Reviews · NeurIPS 2020]

Review 1

Summary and Contributions: The authors approach the problem of optimizing autoregressive spike train models for modeling neural spike trains in response to stimuli. They suggest that a focus on optimizing these autoregressive models by maximizing likelihood on one-step prediction often leads to unstable models that don't predict actual spike trains well. This happens because at training the model is constrained on actually observed spike train values at time steps <t to predict tilmestep t (data-conditioned), but in prediction time, it is only conditioned on it's own prediction at time steps <t (free-running). This can lead to instability in certain regimes. They take inspiration from the GAN literature to rethink the objective they are actually optimizing for and suggest maximum mean discrepancy, coupled with spike train kernels, as a suitable alternative to MLE and claim to show that it outperforms MLE.

Strengths: Updated Review: I'd like to thank the authors for answering my concerns re: the stochastic nature of the kernel optimization. I am now even more confident in my assessment that this is a good submission and an accept from me. In response to an under constrained general MLE framework for fitting autoregressive models, the authors introduce a method to balance multiple objectives (fidelity to outputs under data-constrained and free-running conditions). They achieve this with model-based MMD (sometimes with the addition of a likelihood objective), which requires matching of free-running and data-constrained model features, essentially maximizing likelihood under the constraint of requiring faithful behavior in both conditions. This leads to a model that is both faithful and stable in the free-running condition. The results show that their method comes close to capturing the MLE solution for a simple model built on top of toy data with no history dependence in finite samples, and improves upon the MLE model for toy data with explicit history dependence. They verify this on real datasets from monkey and human cortex and show similar improvement over MLE fits. I think this work has a general significance and relevance to people in the neuroscience community at NeurIPS, but also for people modeling other scenarios with point process dynamics with an emphasis on explainability in their models.

Weaknesses: The authors discuss the the effects of different kernels, and also on the stochastic nature of their optimization procedure, but don't show work leveraging the stochastic nature of their procedure to obtain the expected values of many samples of this optimization process. It's difficult to determine if the differences in kernel estimates fo the history dependence function are consequences of the kernels used or of the stochastic nature of the optimization without this.

Correctness: On page 7 you wrote : "We used a dataset previously used in Hocker and Park [21] and (original reference?)" please add original reference.

Clarity: Yes, the paper is well written and clear.

Relation to Prior Work: Previous work is discussed clearly, and it is clearly demonstrated how this work is a novel contribution.

Reproducibility: Yes

Additional Feedback:


Review 2

Summary and Contributions: The authors address the important problem of run-away excitation in GLM models, a common approach to model the dynamics in neuroscience. Typical estimation method maximizes the data-conditioned log-likelihood, i.e., maximize the probability of observing the data at the next time step given past observations. However, this might not necessarily maximize the free-running likelihood, i.e., recursively using samples generated by the model itself to predict accurate samples for future. Directly maximizing the free-running likelihood from the data is hard. Hence, the authors propose to regularize the traditional data-conditioned likelihood maximization with an additional loss: minimize a Kernel based distance between the model generated spike trains and the data. They present a score-function based estimation method for arbitrary kernels but with higher variance, and an alternate model-based kernel approach that has lower variance.

Strengths: Soundness of Claims The approach is well grounded in the widely used Kernel methods. Significance The authors present a viable method to fix the run-away excitation problem in GLMs. The presented estimation methods that can be applied in different conditions and under different assumptions. As shown by the analysis using autocorrelation, the Kernels can incorporate different features of interest in the responses. Novelty This work presents a different solution to the common problem of run-away excitation in GLMs. Relevance The topic is very relevant to GLMs, a popular modeling tool in Computational Neuroscience, and many in NeurIPS community will find it useful.

Weaknesses: Soundness of Claims As described in more detail below, statistical significance is unclear for most of the analyses. Novelty Quantitative comparison of this approach to existing methods for this problem is absent. Perhaps a simple regularization on history filter would have fixed the problem?

Correctness: Majority of the discussion in Sec 4.3 (Lines 237-250) relies on observations in one model fit for kernel type in Figure 4. However, the lack of error bars in Figure 4 makes the conclusions uncertain. All panels in Figure 4 must have error bars. For example, are the differences in history filter in Figure 4A consistent across multiple fits of model and across different partitions of data? Same for other panels. Many of the spike-train statistics computed on ML fits could be dominated by the few trials that show run-away activation (For example, in Figures 3E, J). What if we just identify and ignore the trials with run-away activation? Does this common (but naive) approach fix this problem? ------ Update post review: The authors do a reasonably good job addressing the concerns. Addition of error-bars in estimated fits and statistics is very helpful, and should go in the final paper. Increasing the score upwards.

Clarity: Section 2.3 and Line 123 describe minimization of dMMD(q_\theta, p_\theta), but Section 3 and results describe minimization of dMMD(\hat{p}, p_\thetha) (See line Eq 10-16, for example). Describe all the Kernels used in the Results (sec 4.2, 4.3) in the Methods (sec 3.2). Small comments: -- Explain in more detail why minimization of KL divergence with free-running likelihood is hard (line 100). -- Line 117: this -> these -- Line 122 : to -> too -- Line 187 : Give figure number. -- Line 192 : Figure 2 -> Figure 2F -- Lines 197 - 211 : Figure 3 (and its panels) must be referred to in main text. -- Line 201, 216 : Mention how is \alpha chosen. -- Line 214 : Fix [original reference]

Relation to Prior Work: Yes, multiple prior works have been mentioned. A related recent work could be included in the paper: https://doi.org/10.1101/2020.06.11.145904

Reproducibility: Yes

Additional Feedback:


Review 3

Summary and Contributions: They optimized many steps of an auto-regressive model for spike trains rather than just one step, using MMD. This avoided instability during generation that is common in models that don't do this.

Strengths: Seems like an important advance.

Weaknesses: It wasn't clear whether it could be used at scale. But it wasn't clear it couldn't, so this is not much of a weakness. Update post-author-response: Given the authors response, it looks like this doesn't have the best scaling with the number of spikes. But it's not so bad that it will kill the methods. So my score won't change.

Correctness: I think so.

Clarity: Yes! Both on an absolute scale, and the best written paper of the six I reviewed.

Relation to Prior Work: I think so, but I'm not really an expert.

Reproducibility: Yes

Additional Feedback: None.

[Author Response · NeurIPS 2020]

We thank the reviewers for their very useful comments. We are encouraged by reviewers thinking our work "is a novel
contribution" (R1), its topic "is very relevant to GLMs" (R2) and "seems like an important advance" (R3). We are glad
they thought our work has general significance in the neuroscience community at NeurIPS (R1) and many will find it
useful (R2). We address the reviewers' comments below and we will incorporate all the feedback. Rebuttal Figure 1
shows some selected panels (for space reasons) from Figs. 3 and 4 that we hope illustrate how changes proposed by the
reviewers will be incorporated. We will incorporate error bars by repeating the optimization procedure multiple times
and we will add L2 regularization on the history filter to compare with our method.

**R2 suggested regularizing the history filter.** As R2 noted, shrinkage of the history filter can lead to improved stability
of the model. Rebuttal Figure 1 shows results for L2 regularization on the history filter. Results suggest that increasing $\alpha$
in L2 regularization to match the data firing rate or stabilize the model (see next response for the choice of regularization
strength) in general leads to worse general performance than MMD regularization. In contrast, MMD regularization
can be used to explicitly match any quantity and not just reduce the size of the history filter coefficients as in L2
regularization. MMD regularization yielded parameters with higher likelihood values (Rebuttal Figure 1 D, F) that also
captured better the data autocorrelation (Rebuttal Figure 1 B, I) (data autocorrelation was smoothed to improve visuals).
Note that the (model-based) MMD in Rebuttal Figure 1C is smaller for L2 regularization than for MMD regularization,
but this is simply due to the L2 regularized model matching its free-running and data-conditioned distributions.

**Choosing regularizer weight ($\alpha$ in NLL + $\alpha$MMD) (R2).** There are multiple possible strategies for choosing the
hyperparameter $\alpha$—one could focus on matching one of the key statistics of the free-running model (e.g. mean firing
rate) with the data, or minimize the runaway excitation. For Figure 3 Monkey-PMv dataset we performed a grid search
and chose the smallest $\alpha$ value such that the mean firing rate of 8000 samples differed less than 10% from the mean
data firing rate. We used the same criteria to determine the regularizing weight in the added L2 regularization (Rebuttal
Figure 1 A-D). For Figure 3 Human Ctx dataset we chose the smallest $\alpha$ such that out of 2400 samples none showed a
diverging firing rate. For Rebuttal Figure 1 E-I we chose the smallest $\alpha$ such that out of 8000 samples none showed a
diverging firing rate. We would state in the final version how we choose $\alpha$ for all the models in Figure 4.

**Statistical significance and error bars (R1, R2).** As pointed out by R1 and R2, our estimation procedure is stochastic.
Rebuttal Figure 1 shows panels from Figures 3 (Rebuttal Figure 1 A-D) and 4 (Rebuttal Figure 1 E-I) with the results of
repeating the optimization procedure 20 times. For Figure 4 panels, we illustrate variability for one choice of kernel
(same model-based kernel as in Figure 3, autocorrelation of the history filter convolved with spikes). The shading in
the curves and the error bars in the barplots represent the 95% percentiles of the distribution obtained by repeating
the optimization. The lines and bar heights represent the mean over optimization repetitions. We can see that for the
examples shown here the stochastic optimization with the proposed MLE initialization is robust and similar parameters
are found consistently. Similar robustness was observed when repeating the optimization procedure multiple times for
the Human Cortex dataset of Figure 3 (not shown but would be incorporated in the final version).

**Scaling (R3).** Our procedure is definitely more expensive than MLE as it involves sampling and computing MMD
during the optimization—in fact, we initialize the optimizer at the MLE. The time and space complexity varies with
many factors; e.g. score function or model-based MMD, biased or unbiased estimator and the choice of the kernel—for
some kernels, it is quadratic in the number of spikes, and for some other kernels, it only depends on the duration of
recording. All the optimizations in the submission take as much as a minute to five in a desktop computer (compared
with seconds for MLE) and there is room for improvement in the current implementation of the procedure. The authors
have experiences with designing computationally efficient kernel and optimization tricks, and we plan to continue to
improve the practicality of this novel concept.

**Generating from ML fit discarding run-away excitation trials (R2).** Removing "outlier" trials from the generative
model has several disadvantages: (1) it can be very inefficient, (2) ML cannot guarantee that the truncation will result in
a "good" distribution, and (3) interpretation of the model parameters is no longer straightforward.

Rebuttal Figure 1: **(A) Corresponds to 3B.. (B) Corresponds to 3C. (C, D) Correspond to 3D. (E) Corresponds to 4A. (F) Corresponds to 4B. (G) Corresponds to 4C. (H) Corresponds to 4D. (I) Corresponds to 4H.**

[Meta-Review · NeurIPS 2020]

In this submission, the authors attack the problem of modeling spike trains in neural data using auto-regressive GLM models. The authors recognize a disparity in the set up for training a MLE estimate of the model parameters and a “free running” model used for inference as has been observed in training RNNs. This disparity may lead to unnaturally long sequences and consequently in spike trains, runaway excitation in the spike train history. To address this issue, the authors propose a new method for fitting GLM’s based on maximum mean discrepancy, coupled with spike train kernels. The authors show favorable predictive performance with respect to MLE methods on real and synthetic neural data. All reviewers found the method novel, the experiments appropriate and the presentation of the methods extremely clear. For these reasons, this paper shall be published at NeurIPS.